# An Intelligent System for Automatic Selection of DC-DC Converter Topology with Optimal Design

**Song Wang[1*], Yi Lu Murphey[1*], Wencong Su[1], Mengqi Wang[1], Van-Hai Bui[1], Fangyuan Chang[1],**
**Can Huang[2], Lingxiao Xue[3], Felipe Leno Da Silva[2], Ruben Glatt[2]**

[1]University of Michigan-Dearborn

[2]Lawrence LivermoreNational Laboratory

[3]Oak Ridge National Laboratory

[1]{songwan, yilu, wencong, mengqiw}@umich.edu

[2] {huang38,lenodasilva1,glatt1}@llnl.gov

[3]xuel@ornl.gov

## Abstract

In this paper, we present an intelligent system that has the capabilities of automatically selecting topology classes and optimizing circuit parameters of DC-DC power converters for a given design specification. The system, Machine-Learning-enhanced Automated Circuit Configuration and Evaluation of Power Converters (ML-ACCEPT), uses a hybrid of machine learning technologies, decision tree inference, reinforcement learning and deep neural networks. The system gives high accurate recommendations of design topology classes and computationally efficient results in optimizing power efficiencies in power converter design.

## 1 Introduction

Electrical power converters (e.g., AC to DC, DC to DC, DC to AC) are critically important in today's electronic world as it processes over 70% of the electricity usage (Bose 2013). They become even more important in the coming decades for the key role they play in the carbon-neutral energy system (Hannan et al. 2019). According to a recent report (Market Watch, 2021), the power-converter market was valued at 207 million USD and is projected to reach 292 million USD by 2026, at a compound annual growth rate (CAGR) of 5.9% during the forecast period.

The power converters consist of interconnected individual circuit components (e.g., resistors, capacitors, inductors, diodes, and switching devices), making their design quite complex and prone to inefficiency. Some existing design tools have a certain level of intelligence to aid the design process. However, to our best knowledge, none of them has the capability of automating the electrical circuit design processes. The circuit design usually involves selecting, configuring, and optimizing the individual components to enable available sources (e.g., high-voltage DC power) to be converted to the desired output (e.g., lower-voltage DC power with a desired voltage ripple), subject to application-specific (e.g., plasma generation and automotive applications) thermal and packaging considerations. The state-of-the-art circuit design of power converters is still heavily reliant on human experts to manually select the optimal topology and determine the design parameters with human's experience and intuitions, which can be very time-consuming, inefficient, and labor intensive. Designing a converter to meet a specific application requirement is a complicated process due to the wide range of design components, parameters, topologies, and their corresponding performances.

In this paper, we present a machine learning framework, Machine-Learning-enhanced Automated Circuit Configuration and Evaluation of Power Converters (ML-ACCEPT), designed to make intelligent selection of DC-DC converter topologies that satisfy a given design specification and optimize design parameters. This paper is organized as follows. Section 2 presents the major machine learning algorithms in ML-ACCEPT, Section 3 presents the experiments we conducted to evaluate the performances of ML-ACCEPT, and Section 4 concludes the paper.

## 2 Automatic Selection of DC-DC Power Converter Topologies and Design Optimization

Fig. 1 presents an overview of ML-ACCEPT. ML-ACCEPT combines decision tree (DT) learning (Safavian and Landgrebe 1991), reinforcement learning (RL) (Sutton and Barto 2018), deep learning (LeCun, Bengio and Hinton 2015), physics-based simulation, and expert knowledge to make an intelligent recommendation of DC-DC power converter topology classes that meet the specified design

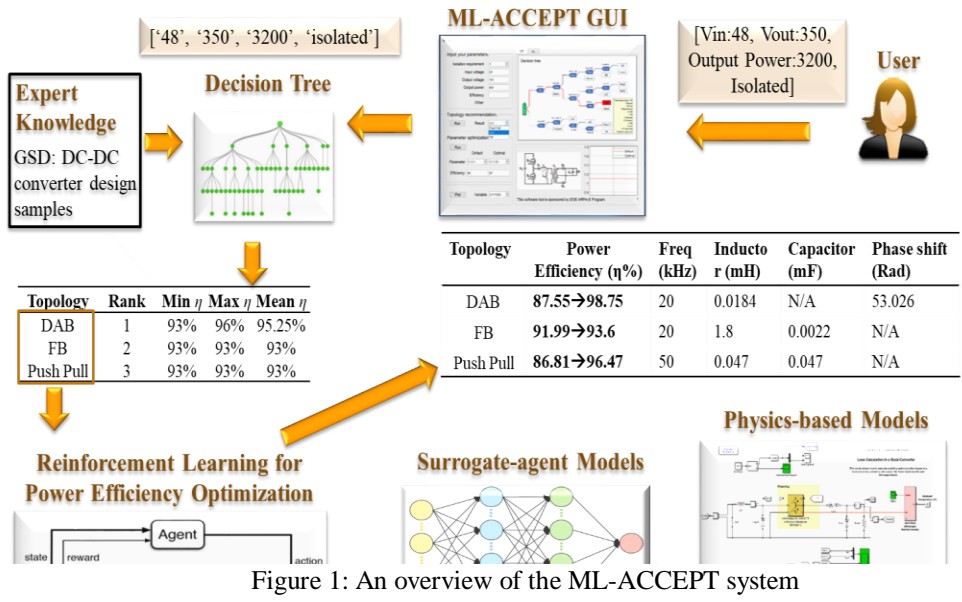

Figure 1: An overview of the ML-ACCEPT system

and performance requirements. The recommendation includes the information of the best topology classes, optimal settings of major design parameters and the expected power efficiency. It first collects a gold standard data (GSD) of DC-DC design samples in 11 topology classes based on expert knowledge. A decision tree is trained on the GSD to make a recommendation of topology classes that best meet the input design specification along with the expected power efficiency performances. Based on the selected topology classes, we use an RL algorithm to search for the optimal design parameters with respect to the power efficiency. RL uses the data generated by the simulation models developed based on the physics of all 11 topology classes for parameter evaluation. In order to reduce the time of online searching and offline training, we developed a deep neural network as the surrogate model for each topology class to mimic the respective physics-based simulation model (PBSM).

## 2.1 Exploring Expert Knowledge for DC-DC Converter Topology Selection

Based on an extensive study of the DC-DC power converters used in engineering practice and research (Zehendner and Ulmann 2017; Falin 2010; Gorij et al. 2019; Paez et al.

selected topology classes and the number of design cases collected by power electronics experts. These data samples are referred to as the Gold Standard Data (GSD) throughout this paper. The design samples were collected from DC-DC converter products (Digi-key, 2021; TI products, 2021), design tutorials provided by electronics vendors (Linear technology, 2021), text books (Hart, D. W. 2011; Mohan, N., Undeland, T.M. and Robbins, W.P. 2003; Zehendner and Ulmann 2017), and technical papers (Falin, J. 2010, Gorij et al. 2019; Paez et al. 2018), and authors' own R&D projects.

## 2.2 A Decision Tree for Automatic Selection of DC-DC Converter Topology

A DT is a nonparametric classification model constructed from a given set of training data and a fixed set of attributes using a recursive search scheme. We use the GSD shown in Table 1 as the training data to build a DT model for automatically selecting the DC/DC converter topologies that meet the design requirements. Five key design attributes were used as input feature vectors: {Input voltage, Output voltage, Output power, Power efficiency, Isolation}. We added an additional constraint attribute x6: Vin/Vout, which reflects Step-down or Step-up property in DC/DC

| Topology | FB | Forward | Push Pull | HB | DAB | LLC | Buck | Boost | Buck-boost | Switch-cap | Flyback | Total |
|---|---|---|---|---|---|---|---|---|---|---|---|---|
| #Samples | 89 | 56 | 85 | 25 | 74 | 102 | 37 | 46 | 41 | 45 | 27 | 627 |

Table 1: GSD: DC-DC converter design samples in 11 commonly used topology classes

2018), we identified 11 classes of commonly used and emerging DC/DC converter topologies and collected design cases to build the knowledge base. Table 1 shows the

converters. The following describes the three major computational components in the DT learning process.

### Building a decision tree for DC-DC converter topology selection using GSD

Let D be all the design cases contained in GSD. The DT learning algorithm is a recursive process that uses the well-known classification and regression tree (CART) method (Safavian, S.R. and Landgrebe, D. 1991) to evaluate and select attributes to split at each branch node. It starts with a single node named as the root with the all the design cases D. During the learning process, in each node a particular subset of D is processed. If all elements of the subset meet with one of the stopping criteria, the node is tagged as a leaf and the split stops. Otherwise, the attribute has the minimum Gini value is chosen to split the node into branches along with the new sub-datasets based on their attribute values. The procedure repeats until one of the following stopping criteria is met, (1) all design cases in the current training set belong to a single topology class, (2) the number of cases associated with a node is less than a predetermined threshold, MinParentSize, and (3) If the node is split, the number of cases in at least one child node would be less than the pre-determined threshold, MinLeafSize.

After the DT learning, each leaf node contains the topology classes of the design cases in GSD that match with the design attributes specified along the path from the root to the leaf node, and the statistics of the power efficiencies associated with the design cases in each topology class.

### Decision Tree Learning from interval samples

Many power converter design applications, users specify attributes such as input and output voltages in the form of intervals rather than single values, which are typically the attribute format used in building a DT. In order to make the DT more powerful, it is important to enable a DT to learn design attributes directly from intervals. We developed a new algorithm, Decision Tree learning from Intervals (DTI), which uses a modified C4.5 algorithm. In the C4.5 algorithm, a probability is calculated to determine splitting attribute and splitting criterion whereas in DTI we replaced the probability measure with probabilistic cardinality (PC) (Qin, Xia and Li 2009). The PC of the training dataset over an interval [a, b] is the sum of the probabilities of each instance whose corresponding attributes falls in the interval [a, b]. The DTI algorithm can generate DTs from both interval samples and single value samples; therefore it is more flexible in dealing with various training data formats.

### Parameter optimization

As we discussed above the two parameters, MinLeafSize and MinParentSize control the size of DT. In order to avoid DT from overfitting the training data, we developed following approach to select the proper values of these two control parameters. A step-by-step grid search with a search range and a step size assigned to parameter MinLeafSize and to MinParentSize. For each pair of values of these two parameters, we conduct a 10-fold cross validation strategy to generate and evaluate the decision tree models using this pair of parameters values. With the 10-fold cross validation, the training data are randomly sampled into stratified 10 partitions among all 11 topology classes. At each fold, a decision tree is trained using 9 partitions of the data samples using the pair of parameters, and evaluated on data in the remaining partition. The pair of values that give the best average performances over all 10 folds are chosen to be the optimal values of MinParentSize and MinLeafSize respectively for use in the DT learning process.

## 2.3 Developing RL-based algorithms for circuit design parameter optimization

As discussed in section 2.2, the DT is designed to provide feasible DC-DC converter topology classes that meet with the input design requirement. The next task is to optimize the circuit design parameters of each candidate topology with respect to power efficiency using RL (Glatt et al. 2022). For a finite Markov decision process (Puterman 2014), Q-learning (Watkins and Dayan 1992) finds an optimal policy in the sense of maximizing the expected value of the total reward over all successive steps, starting from the current state. We also investigated the deep deterministic policy gradient (DDPG) method (Lillicrap 2015) to optimize the performance of converters. DDPG is an off-policy deep reinforcement learning (DRL) algorithm aimed at learning dynamics in continuous state and action spaces. It is a more generalized training approach with deep neural networks. Both Q-learning and DDPG can work well if the number of optimizing parameters is small, but DDPG performs better for problems with large state/action spaces.

During the training process, the learning agent in both methods interacts frequently and directly with the dynamic environment by running the PBSM. Since a single simulation request can take from several seconds to tens of seconds to complete, the training time of the RL/DDPG-based optimization system is too long to be feasible for this application. In order to reduce the training time for RL methods, we developed a surrogate model for each topology class of the power converters. The surrogate model is a deep neural network (DNN) trained using the data generated by the PBSM to mimic its behavior.

The circuit design parameters being used to optimize power efficiency include, switching frequency, inductance, capacitance, input voltage, output voltage, output power, phase shift in DAB converter, and other parameters related to MOSFET (resistance, diode voltage, rising and falling times). Each of these parameters has a pre-defined search

| Optimization Method | Performance (power efficiency) | Computational cost for $N$=200 different design cases of {Vin, Vout, Pout} | | |
|---|---|---|---|---|
| | | Online Optimization | Offline Surrogate Modeling | Offline Pre-training |
| Q-learning | [0.8708, 0.9830] | <1 second | ~24 hours | $T_Q > 2 \times T_{DDPG}$ |
| DDPG | [0.8704, 0.9805] | <1 second | ~24 hours | $T_{DDPG} \in [2,3]$ hours |
| Brute force | [0.8708, 0.9830] | $N$ x (3 to 5 hours) | N/A | N/A |

Table 2: Comparison of computational cost in topology parameter optimization.

range and grid size. For each DC-DC converter topology class, systematically run its PBSM at every grid point in the parameter space to generate measures of power efficiency. The dataset is used to train a deep neural network (DNN) as surrogate model of the PBSM for the topology class. The DNN is then used by the RL algorithms to estimate the power efficiency of a DC-DC converter with any sample of design parameters in the defined space.

## 3 Experiments

In this section, we present experiments conducted for evaluating the performances of ML-ACCEPT. We trained a DT using the GSD (see table 1) combined with a set of 104 design cases with attributes specified in intervals. During the DT learning, the two tree pruning control parameters, MinLeafSize and MinParentSize were set to 2 and 5 respectively, which were determined by the algorithm presented in Section 2.2. We evaluated the performances of the DT models using hit rates (HR) generated on the test sets in a 10-fold cross validation procedure. The HR was over 89%, which implies that more than 89% of the test cases' ground truths were contained in the DT selected topology classes that satisfy the user's specified design criteria.

The following provides a comparison of the design cycles by human-experts and the ML-ACCEPT software in terms of time cost. For each design case submitted by a user, the DT takes less than 1 second to generate a list of recommended topology classes that meet the specified design requirement. Table 2 summarizes the computational cost for offline training and online optimization operations of Q-learning, DDPG and Brute force methods when applied to 200 design cases.

The Brute force method tests all possible solutions for each design case and selects the best solution. For the 200 design cases, this method took more than 600 to 1000 hours to complete, about 3 to 5 hours per case in average. In the implementation of the Q-learning method, due to the restriction of the size of the Q-table, the 200 design cases were evenly divided into two batches, and the same offline pre-training process is applied to each training batch. For the DDPG method, all design cases can be trained simultaneously and only one offline pre-training process is required for all design cases. The offline pre-training process is to optimize the weights of DDPG, which took 2

to 3 hours. The off-line pre-training time for Q-learning is more than twice of the time needed for training DDPG.

Both Q-learning and DDPG use the surrogate model, DNN, to mimic the PBSM for each topology class. PBSMs were all implemented using Matlab Simulink. The DNN has two hidden layers with 512 nodes in each hidden layer, and its loss function is based on the mean square error. Its training data, 30,000 data points, were generated by running the respective the PMSM model of the topology class repeatedly using sampled parameter values This dataset is then randomly partitioned into a training (70%) and a test set (30%). The training of DNN took 1,000 training epochs, and the average prediction error of the DNNs is about 0.013 on the test data. The process for surrogate modeling and training took approximately 24 hours for each topology class. After the training process, the optimal DNN is used by Q-learning and DDPG to predict the power efficiency of a power converter for a given instance of the design parameters. After the pre-training and the use of the surrogate model, the online design parameter optimization processing by either Q-learning or DDPG is very fast, less than 1 second for all 200 design cases.

Based on our survey data, typically a human-expert design cycle can take at least several days to complete the entire process, including information gathering, selecting topology candidates, developing the simulation models for the selected topologies, trying and evaluating various design parameters, and running various experiments. ML-ACCEPT gives accurate selection of DC-DC converter topology classes and takes less than 1 second to conduct design parameter optimization.

## 4 Conclusion

We have presented a machine learning framework, ML-ACCEPT for building an intelligent system for automatic selection of DC-DC converter topologies and optimization of design parameters. ML-ACCEPT is built on the design samples collected based on expert knowledge and uses DT models for automatic selection of DC/DC converter topologies that meet the design requirements. Our experiment results show that the HR of the DC-DC topology selection by the DT has reached more than 89%. ML-ACCEPT implemented and evaluated two RL-based algorithms, Q-learning and DDGP, for optimizing design parameters with respect to power efficiency for a given DC-

DC converter topology with specified design descriptions. The experimental results demonstrated that when combined with a surrogate model of topology simulation, both RL algorithms took less than a second in obtaining optimal design parameters. In conclusion, ML-ACCEPT provides an intelligent software tool for DC-DC converter design that can perform accurately on topology selection and is highly efficient in design optimization.

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
