# OpenReview forum: "An Intelligent System for Automatic Selection of DC-DC Converter Topology with Optimal Design"
_AAAI.org/2022/Workshop/ADAM — AAAI 2022 Workshop ADAM_

### Official Review · Reviewer_rRz5 · 2021-11-29
**Optimizing DC-DC converter topology using ML, lacks related work**

**Rating:** 6
**Confidence:** 4

**Review:**

This paper presents a framework to optimize the design of DC-DC converters using machine learning methods. A decision tree is first used to identify the topology class of a given set of design features that is trained on a set of expert chosen topologies, then a reinforcement learning algorithm is used to optimize the design of the circuit for that topology class. To speed up the RL part, a neural network is trained as a surrogate for the reward function that is trained using power efficiency estimates from a physics based simulation/calculation. The method is shown to outperform brute force search for topology and circuit parameters.

The application and framework is sound, however there is no discussion of related work and comparison to any baselines. Also, it would be helpful to justify why RL is needed rather than an optimization approach such as Bayesian optimization (i.e. what causes the environment to be dynamic), also the neural network is not really physics informed - its simply trained on data from physical simulations.

---

### Official Review · Reviewer_oMhS · 2021-11-29
**ML framework for DC-DC converter design**

**Rating:** 6
**Confidence:** 4

**Review:**

This paper proposes a two stage ML framework for designing selecting topology classes and optimizing circuit parameters of DC-DC power converters for a given design specification. The first stage involves designing a Decision Tree for selecting a topology class for the converter, while the second stage involves a RL agent for further design exploration. As done in most RL-based design framework, a surrogate model is built for fast function evaluation in the RL framework. While there is no novelty in the ML aspect, but the empirical results show that this framework may be useful for the application domain. The paper lacks in providing some critical details such as the aspect of reward engineering for the RL agent as well as the literature review is quite minimal and does not highlight many relevant work in the areas of RL-based design and use of ML-based surrogates for design optimization.